# The Impact of the Association between Cancer and Diabetes Mellitus on Mortality

**DOI:** 10.3390/jpm12071099

**Published:** 2022-07-01

**Authors:** Sung-Soo Kim, Hun-Sung Kim

**Affiliations:** 1Department of Health Administration & Healthcare, Cheongju University, Cheongju 28503, Korea; mra7033@naver.com; 2Division of Endocrinology and Metabolism, Department of Internal Medicine, Seoul St. Mary’s Hospital, College of Medicine, The Catholic University of Korea, Seoul 06591, Korea; 3Department of Medical Informatics, College of Medicine, The Catholic University of Korea, Seoul 06591, Korea

**Keywords:** cancer, diabetes mellitus, hypertension, association rules mining

## Abstract

The prevalence of cancer, diabetes mellitus (DM), and hypertension is increasing in ageing populations. We analyzed the association of DM with cancer and its effects on cancer mortality. The data of 2009–2018 from the Korea National Hospital Discharge In-depth Injury Survey were used; 169,959 adults with cancer as the main diagnosis were identified. The association rule for unsupervised machine learning was used. Association rule mining was used to analyze the association between the diseases. Logistic regression was performed to determine the effects of DM on cancer mortality. DM prevalence was 12.9%. Cancers with high DM prevalence were pancreatic (29.9%), bile duct (22.7%), liver (21.4%), gallbladder (15.5%), and lung cancers (15.4%). Cancers with high hypertension prevalence were bile duct (31.4%), ureter (30.5%), kidney (29.5%), pancreatic (28.1%), and bladder cancers (27.5%). The bidirectional association between DM and hypertension in cancer was the strongest (lift = 2.629, interest support [IS] scale = 0.426), followed by that between lung cancer and hypertension (lift = 1.280, IS scale = 0.204), liver cancer and DM (lift = 1.658, IS scale = 0.204), hypertension and liver cancer and DM (lift = 3.363, IS scale = 0.197), colorectal cancer and hypertension (lift = 1.133, IS scale = 0.180), and gastric cancer and hypertension (lift = 1.072, IS scale = 0.175). DM increased liver cancer mortality (*p* = 0.000), while hypertension significantly increased the mortality rate of stomach, colorectal, liver, and lung cancers. Our study confirmed the association between cancer and DM. Consequently, a patient management strategy with presumptive diagnostic ability for DM and hypertension is required to decrease cancer mortality rates.

## 1. Introduction

With the recent advances in science and technology, the early diagnosis of cancer has become possible, and there is growing interest in the management of long-term cancer survivors [1,2]. With the increased long-term survival rate of patients with cancer, their ageing process has become characteristic, as various chronic diseases such as diabetes and hypertension, and cardiovascular diseases are becoming important issues among cancer survivors [3,4]. Many studies have reported diabetes and hypertension as the most common chronic diseases among cancer survivors. Particularly, diabetes is an independent risk factor for cancer [5,6]. Moreover, the incidence of diabetes is high, even in patients with cancer [7]. After all, diabetes is closely related to cancerogenesis, which requires more attention because of the increased risk for cardiovascular disease in patients with cancer.

Diabetes plays an important role in cancer progression and cancer incidence [8,9]. Since diabetes and hypertension are well-known prognostic factors for the death of patients with cancer, the interest in cancer prevention or early detection is increasing [10,11]. Particularly, the incidence of liver, lung, colorectal, and stomach cancers was high among male Asian patients with diabetes [4], and physicians treating patients with type 2 diabetes in clinical practice should manage patients with cancer risk factors more closely. Although there are many retrospective studies on the correlation between cancer and diabetes, there are few studies confirming the additional association from cancer onset to death. Therefore, this study aimed to analyze the prevalence of diabetes mellitus in patients with cancer, to determine the association between diabetes mellitus and cancer and its impact on mortality.

## 2. Materials and Methods

### 2.1. Study Population

This study used data from the Korea National Hospital Discharge In-depth Injury Survey (KNHDIS) over 10 years (2009–2018). The KNHDIS is a nationally approved survey (Statistics Korea Approval No. 117060), which entails the annual collection of information of patients discharged during the previous year by the Korea Disease Control and Prevention Agency (KDCA) [12,13]. According to the data provision procedure of the KDCA, personally identifiable data were excluded. The study participants were adults (≥19 years) who were discharged from medical institutions with more than 500 hospital beds. The data of 169,959 patients with a principal diagnosis of C00–C97 (codes for malignant neoplasms in the International Classification of Diseases, 10th Revision [ICD-10]) were finally collected and analyzed.

### 2.2. Variables and Measures

The participants’ general characteristics included sex, age, insurance type, admission route, and treatment outcome. Age was classified into four groups (19–44 years, 45–64 years, 65–74 years, and ≥75 years). Insurance types were classified into national health, Medicaid I, Medicaid II, and others. Hospitalization routes were categorized into emergency, outpatient, and others. Treatment outcomes were classified as improved, not improved, death, and others. The clinical characteristics of the study participants included surgery, chemotherapy, radiotherapy, hyperthyroidism, and hypothyroidism. These variables were classified based on the ICD-10 and the International Classification of Diseases, Ninth Revision, and Clinical Modification (ICD-9-CM). If the main surgery code of a patient was not null, the patient was classified as having undergone surgery. Data on chemotherapy (ICD-9-CM codes: 99.25, ICD-10 codes: Z51.1), radiotherapy (ICD-9-CM codes: 92.21–9227 or 9229), hyperthyroidism (ICD-10 codes: E05.0–E05.9), and hypothyroidism (ICD-10 codes: E02–E03.9, E89.0) were extracted. Diabetes mellitus (ICD-10 codes: E11.0–E14.9) and hypertension (ICD-10 codes: I10) were the main variables. Using the ICD-10 codes, the cancer-involved organ was identified. For example, stomach cancer, colorectal cancer, liver cancer, and lung cancer were classified as C16.0–C16.9, C18.0–C19, C22.0–C22.9, and C33–C34.9, respectively.

### 2.3. Statistical Analysis

A frequency analysis was conducted to understand the prevalence of diabetes mellitus and hypertension in the study participants. General characteristics, diabetes, and hypertension were compared between groups using the χ^2^ test applied Bonferroni correction. This study analyzed the prevalence of diabetes mellitus and hypertension in patients with major cancer types; the χ^2^ test was used to assess the difference between their distributions. Association rule mining (ARM) was used to analyze the association rule between cancer, diabetes, and hypertension. The Apriori algorithm was used to analyze the association rules. The Apriori algorithm is an unsupervised machine learning model and is designed to focus on the occurrence of events mainly used in association rules [14]. Its application has been expanded to the medical field beyond the marketing field because it is effective in identifying patterns between diseases [15,16,17,18]. Support, confidence, and lift are the evaluation criteria of the Apriori algorithm [19,20]. In this study, support refers to the ratio of patients diagnosed with both diseases A and B among the study participants. Confidence refers to the ratio of patients who had been diagnosed with disease A, and then became associated with disease B. Lift refers to the probability of diagnosing diseases A and B together, compared to the probability of diagnosing diseases A or B independently. Although lift was the main index presenting the strength of the association rule, this study used interest support (IS), which considered both support and lift [21]. The formula for each index is as follows Equations (1)–(4).
(1)$$\mathrm{Support}\left(A\to B\right)=\frac{Number\;of\;patients\;with\;disease\;A\;and\;disease\;B}{Total\;number\;of\;patients}$$
(2)$$\mathrm{Confidence}\left(A\to B\right)=\frac{Number\;of\;patients\;with\;disease\;A\;and\;disease\;B}{Number\;of\;patients\;with\;disease\;A}$$
(3)$$\mathrm{Lift}\left(A\to B\right)=\frac{Support\left(disease\;A\;\to \;disease\;B\right)}{P\left(\mathrm{disease}\;A\right)\times P\left(\mathrm{disease}\;B\right)}$$
(4)$$\mathrm{IS}\left(A\to B\right)=\sqrt{\mathrm{Support}\left(A\to B\right)\times \mathrm{Lift}\left(A\to B\right)}$$

After reviewing previous studies, primary association rules that satisfied “minimum support >0.01” and “minimum reliability >0.1” were extracted while considering the study’s characteristics [22]. Coincidences or unrelated patterns (lift ≤ 1) were excluded [23]. The extracted final pattern was visualized as a network graph using the arulesViz library of R [24]. Lastly, a logistic regression analysis was performed after controlling the effects of demographic characteristic variables to understand the effects of diabetes on mortality for the major cancers found in the association rule. All statistical analyses were performed using R version 4.1.0 (R Foundation for Statistical Computing, Vienna, Austria).

## 3. Results

### 3.1. Prevalence of Diabetes Mellitus and Hypertension in the Study Population

The demographic characteristics of the participants and the prevalence of diabetes mellitus and hypertension are shown in Table 1.

There were more male (53.9%) than female (46.1%) patients. The prevalence of hypertension (20.4%) was higher than that of diabetes (12.9%) among the patients with cancer. The prevalence of diabetes mellitus and hypertension was significantly higher in male than in female patients (*p* < 0.001). Regarding the age distribution, 44.6%, 25.7%, and 15.2% of the study participants were aged 45–64, 65–74, and ≥75 years, respectively. The 65–74 age group had the highest prevalence of diabetes mellitus (18.4%), while the ≥75 group had the highest prevalence of hypertension (33.6%) (*p* < 0.001); these differences were significant. The most common treatment outcome was “improved” (88.8%), followed by death (6.1%). The prevalence of diabetes mellitus and hypertension according to treatment outcome was not significantly different. A proportion of 40.7% of the participants underwent surgery. The prevalence of diabetes mellitus (15.2%) and hypertension (21.7%) among the patients who did not undergo surgery was significantly higher (*p* < 0.001). A proportion of 11.4% of the patients received chemotherapy during hospitalization. The prevalence of diabetes mellitus among patients who received anticancer treatments (14.2%) was significantly higher than that among patients who did not receive them (12.7%). However, the prevalence of hypertension was similar, regardless of anticancer treatments. It was found that 0.4% of the patients received radiotherapy, and radiotherapy did not affect the prevalence of diabetes mellitus and hypertension. A proportion of 0.2% of the participants were diagnosed with hyperthyroidism during hospitalization, and patients diagnosed with hyperthyroidism had a higher prevalence of diabetes mellitus (20.3%) and hypertension (26.4%) than those without hyperthyroidism (*p* < 0.001). The prevalence of hypothyroidism was 1.1%, and the prevalence of diabetes mellitus (17.0%) and hypertension (26.4%) was significantly higher among patients with hypothyroidism than among those without hypothyroidism (*p* < 0.001).

### 3.2. Prevalence of Diabetes Mellitus and Hypertension in Patients with Major Cancers

The distribution of major cancers and the prevalence of diabetes mellitus and hypertension among the study participants are shown in Table 2.

The most common cancer was stomach cancer (13.1%), followed by lung cancer (12.5%), colorectal cancer (12.4%), liver cancer (11.7%), and pancreatic cancer (3.2%). Cancer was found in 52.9% of the study participants. Diabetes mellitus was diagnosed in patients with pancreatic cancer (29.9%), bile duct cancer (22.7%), liver cancer (21.4%), gallbladder cancer (15.5%), and lung cancer (15.4%); the differences between these patients were significant (*p* < 0.001). Regarding the prevalence of hypertension, it was highest in patients with bile duct cancer (31.4%), followed by those with ureter cancer (30.5%), kidney cancer (29.5%), pancreatic cancer (28.1%), and bladder cancer (27.5%); the differences were significant (*p* < 0.001).

### 3.3. Association Rule Mining among Diseases

This study analyzed the association rules between cancer, diabetes mellitus, hypertension, hyperthyroidism, and hypothyroidism among the study participants. The strength of the association rules in an increasing order using the IS scale, which was based on support and lift, is shown in Table 3.

Twenty-two association rules in this study satisfied the minimum criteria. The bidirectional association between hypertension and diabetes mellitus was the strongest among the patients with cancer (lift = 2.629, IS scale = 0.426). This suggested that the probability of patients with hypertension being diagnosed with diabetes mellitus was 2.629 times higher than that of patients with cancer being diagnosed with diabetes mellitus; the reverse path has the same interpretation. A bidirectional association between lung cancer and hypertension was also found (lift = 1.280, IS scale = 0.204). This suggested that the probability of patients with lung cancer being diagnosed with hypertension was 1.280 times higher than that of patients with cancer being diagnosed with hypertension. Since it was bidirectional, the reverse path also has the same interpretation. A bidirectional association between liver cancer and diabetes mellitus was also observed (lift = 1.658, IS scale = 0.204). This suggests that the probability of patients with liver cancer being diagnosed with diabetes mellitus is 1.658 times higher than that of patients with cancer being diagnosed with diabetes mellitus; the reverse path has the same interpretation. The next bidirectional association was that between hypertension and liver cancer and diabetes mellitus (lift = 3.363, IS scale = 0.197). This suggests that the probability of patients with both hypertension and liver cancer being diagnosed with diabetes mellitus was 3.363 times higher than that of patients with cancer being diagnosed with diabetes mellitus. A bidirectional association between colorectal cancer and hypertension was also observed (lift = 1.133, IS scale = 0.180). This suggests that the probability of patients with colorectal cancer being diagnosed with hypertension was 0.133 times that of patients with cancer being diagnosed with hypertension. Next, the bidirectional association between gastric cancer and hypertension was analyzed (lift = 1.072, IS scale = 0.175), suggesting that the probability of patients with gastric cancer being diagnosed with hypertension was 0.072 times that of patients with cancer being diagnosed with hypertension. The other association rules follow the same interpretation rule used in the aforementioned ones. In summary, among the cancer types, lung cancer, liver cancer, colorectal cancer, and gastric cancer formed association pathways with diabetes mellitus and hypertension. Hyperthyroidism and hypothyroidism were also included in the analysis. However, they were excluded from the association rule because they did not meet the minimum criteria. Figure 1 shows the network graph visualizing these pathways.

The size refers to support, and the color of the circle means lift. Diabetes mellitus and hypertension are at the center of the graph. It confirms that related cancer diseases in the vicinity form a path with diabetes mellitus and hypertension.

### 3.4. Factors Affecting the Death of Cancer Patients

We previously confirmed that lung cancer, liver cancer, colorectal cancer, and gastric cancer were highly associated with diabetes mellitus and hypertension by conducting an association analysis (Table 3). This study used logistic regression to understand the effects of diabetes mellitus and hypertension on the mortality of these four cancer types. Table 4 shows the analysis results after controlling sex, age, chemotherapy, and radiotherapy variables.

Although hyperthyroidism and hypothyroidism did not appear in the association rules, they were included in the analysis, and their effects on diabetes mellitus and hypertension were determined while controlling for these variables; patients with diabetes mellitus had a lower mortality rate (OR = 0.917, *p* = 0.004). However, patients with hypertension had a higher mortality rate (OR = 1.200, *p* < 0.001). Hypertension increased the mortality of patients with gastric cancer (OR = 1.495, *p* < 0.001). Hypertension also increased the mortality of patients with colorectal cancer (OR = 1.199, *p* = 0.040). Diabetes mellitus (OR = 1.265, *p* < 0.001) and hypertension (OR = 1.242, *p* = 0.001) increased the mortality of patients with liver cancer. Only hypertension (OR = 1.270, *p* < 0.001) increased the mortality of patients with lung cancer. Although hypertension was found to increase the mortality rate of all the four cancer types, diabetes mellitus increased only the mortality rate of patients with liver cancer. Hypothyroidism reduced the mortality in liver cancer (OR = 0.485, *p* = 0.002). Figure 2 shows the results visualized by a forest plot.

## 4. Discussion

As shown in this study, diabetes increases the risk of cancer. It is also known that the incidence of diabetes in patients with cancer is high [4,5,6,7]. Based on previous studies, the risk of various cancers in patients with diabetes is 1.3–2.5 times higher [5,6]. In addition, with the increasing diabetes incidence, the cancer incidence is also expected to increase. As such, patients with diabetes have a high risk of cancer. Therefore, it is expected that special attention should be paid to cancer prevention and management [25].

Although the mechanism of cancer development in patients with diabetes remains unclear, various mechanisms have been proposed. Chronic inflammation caused by hyperglycemia and obesity in patients with diabetes is also known to affect cancerogenesis [26]. Hyperinsulinemia reportedly causes the failure of insulin secretion regulation, and the rapid proliferation and metastasis of cancer cells is promoted by an increase in the biological activity of insulin-like growth factor 1 (IGF-1) [8]. In addition, diabetes causes oxidative stress and DNA damage, which increases the risk of cancer by causing various mutations [9]. In addition, high glucose levels interfere with the cell’s repair of damaged DNA [27]. The fact that metformin lowers DNA damage and prevents cancer explains this mechanism properly [28].

Due to these various causes, an increase in cancer incidence and mortality rates are both observed in patients with diabetes. In a previous study, the mortality rate of patients with cancer and concurrent diabetes was 1.2 times that of patients with cancer without diabetes [29]. However, in this study, a logistic regression analysis of the effect of diabetes on mortality in patients with gastric, colorectal, and liver cancers was performed; there was a strong correlation between these cancers and diabetes. In all the patients with cancer, the OR for diabetes was 0.917, indicating that the number of deaths among patients with diabetes was lower. However, in the patients with liver cancer, the patients with diabetes were 1.265 times more likely to die than those without diabetes. These findings varied with the cancer type. Based on our study, attention should also be paid to hepatobiliary cancer among patients with diabetes. According to the American Cancer Society (ACS) and the American Diabetes Association (ADA), the relative risk of liver, pancreatic, and endometrial cancers in patients with diabetes is more than twice that of normal people. According to a study [30], the risk of liver cancer in patients with diabetes is 1.95 times that in the general population, suggesting the possibility of obesity or a fatty liver in addition to hyperinsulinemia.

In a study of 310,000 patients in England, deaths from heart disease or stroke were significantly reduced in patients with diabetes, but there was no significant change in cancer mortality among them [31]. This suggests that cancer deaths in patients with diabetes have received much more attention than cardiovascular deaths. In a study of more than 400,000 patients with cancer in Denmark, all patients with diabetes who were treated with oral hypoglycemic agents or insulin had a higher mortality rate than those without diabetes [10]. Notably, the duration of diabetes mellitus had no effect on cancer mortality [10]. In a previous Asian retrospective study of 770,000 people followed up for 12 years [11], diabetes increased the mortality rate of all cancer types by 26%. Moreover, their study was a large-scale prospective cohort study. However, there are many things to consider in this regard. Since diabetes itself is not an independent cause of cancer-related death but can be accompanied by various chronic diseases, it may have lowered the prognosis of cancer itself. Lifestyle also plays a major role, and it is known that patients with diabetes have more risk factors than the general population, such as smoking, obesity, and poor eating habits [32]. Finally, as the efficacy of drugs for diabetes and hypertension is improving, the number of cardiovascular disease-related deaths is decreasing, but the number of cancer deaths is increasing.

In this study, the cancer with a particularly high diabetes prevalence was pancreatic cancer. Other cancers with a high prevalence were bile duct cancer (22.7%) and hepatic cancer (21.4%). A study on the correlation between diabetes and cancer revealed that pancreatic cancer had the strongest correlation with diabetes [33]. In a 25-year follow-up study, hyperglycemia increased the incidence of pancreatic cancer by more than 2.2 times, 4.9 times in a 17-year prospective follow-up study in France, and 1.9 times in a 24-year follow-up study in Sweden [34,35,36]. In fact, the correlation between diabetes and pancreatic cancer has already been proven in many studies; it has been shown to be associated with pancreatic cancer mortality [37]. This relationship is clarified by previous studies that reported the development of diabetes 1–2 years before pancreatic cancer diagnosis [5,6]. Regarding diabetes mellitus, insulin resistance and excessive insulin secretion activate, which activates IGF-1 receptor signaling, contributed to the development of pancreatic cancer [8]. Therefore, it is recommended that patients with diabetes undergo pancreatic cancer screening. In addition, since gestational diabetes is a well-known risk factor for pancreatic cancer [38,39], periodic screening tests for pancreatic cancer in diabetic patients should be taken seriously in future.

In this study, the prevalence of diabetes was significantly higher in men than in women and in those aged ≥65 years. In a real-world study of 410,000 people in China, the risk of occurrence and cancer type differed with sex in patients with diabetes [4]. The sex difference is explained by the variation in cancer occurrence with sex and the different incidence rates in each study. For example, in a study conducted in China, patients with diabetes were mostly at risk for prostate cancer (males) and nasopharyngeal cancer (females) [4]. In addition, age (old age), sex (male), obesity, lack of exercise, diet, drinking, and smoking were the most common risk factors for both diseases. This may have partially affected the association.

Although there were differences according to cancer type in this study, the correlation between cancer and hypertension appeared to be strong. Cancer and hypertension share important risk factors for their development. Overweight, poor eating habits, low physical activity, diabetes, and dyslipidemia are major factors in the development of cancer and hypertension. However, the association between cancer and hypertension remains controversial [40,41]. This is because whether the occurrence of cancer is related to the antihypertensive drugs rather than hypertension itself is unclear [42]. A large-scale study in Korea reported that patients treated with angiotensin II receptor blockers (ARBs) had a low cancer incidence [43], but a study conducted in Japan reported that ARBs promote tumor angiogenesis [42]. On the other hand, there is no evidence that any type of antihypertensive drug increases the risk of cancer [44]. Therefore, the controversy over hypertension and cancer is expected to continue. In this study, since only the diagnosis of hypertension was investigated, the causal relationship between cancer and hypertension itself or antihypertensive drugs is unclear. A prospective study on this is required.

This study has various limitations due to the characteristics of a retrospective cohort study [45]. First, it is difficult to explain the causal relationship between diabetes and cancer, and only the correlation can be estimated [46]. Since this study contained discharged patients’ information constructed as episodes of the discharge date, there is no identifier that can track a specific patient. Therefore, since the temporal relationship between cancer occurrence and diabetes is unknown, attention should be paid to the interpretation of the causal relationship between the two diseases. The second limitation is the operational definition of diabetes [45]. As of 2018, the Korean Diabetes Association reported that the prevalence of diabetes in adults above 30 years old was 13.8% [47]. In our study, the rate of diabetes among patients with cancer was 12.9%, which may be mistaken for a lower prevalence when compared to patients without cancer. However, this study included adults aged 19 years and above and not only those aged above 30 years. In addition, since the operational definition of diabetes included only the diagnosis name, this difference seems natural, and the overall prevalence seems to be a reasonable value. Finally, the stage of cancer could not be confirmed, and various related variables that could affect the correlation between diabetes and cancer were not included.

## 5. Conclusions

In this study, the strength of the association between cancer and diabetes was confirmed, and 22 association rules were derived in the order of the strength of the association. To overcome the limitations of numerous studies conducted in a single hospital, this study used real-world data at the national level. In addition, to overcome the limitations of classical statistical analyses, this study used the association rule for unsupervised machine learning. Such an attempt will increase the generalizability and validity of the research results. Our findings show a definite increase in the prevalence of diabetes among patients with cancer. The prevention of complications in patients with diabetes is important, but since cancer is the most common cause of death, physicians should manage patients with cancer risk factors closely. In the future, we propose a new strategy for cancer screening in patients with diabetes and a multicenter cohort study that can follow the disease history of individual patients. It is necessary to establish regular cancer screening tests and prevention strategies for patients with type 2 diabetes.

## Figures and Tables

**Figure 1 jpm-12-01099-f001:**
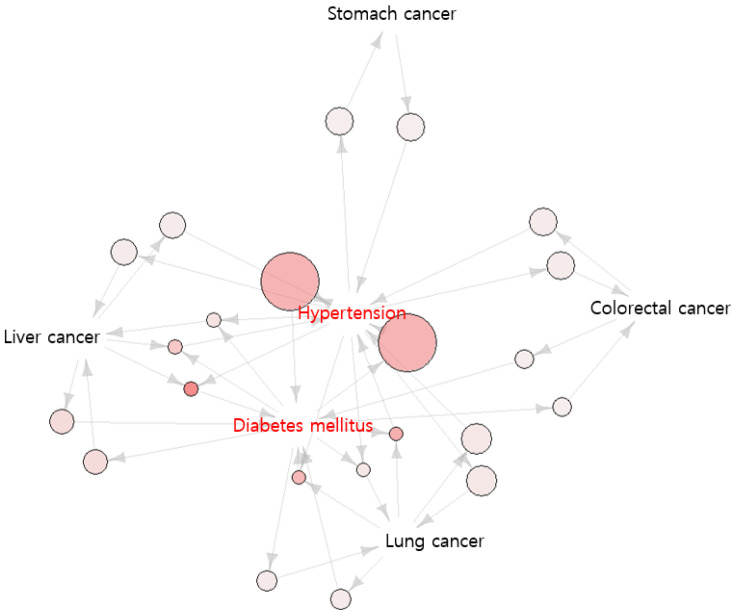
Network analysis for association rule mining.

**Figure 2 jpm-12-01099-f002:**
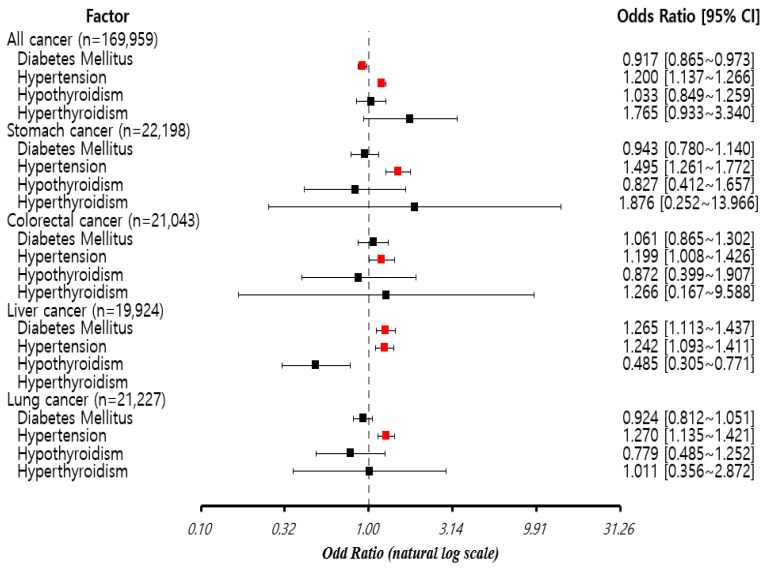
Forest plot of the impact of mortality according to cancer type.

**Table 1 jpm-12-01099-t001:** Baseline characteristics of patients with cancer and concurrent diabetes or hypertension.

Variables	Total *n* (%)	with Diabetes Mellitus		with Hypertension	
		*n* (%)	*p*-Value	*n* (%)	*p*-Value
Sex			<0.001		<0.001
Male	91,526 (53.9)	14,357 (15.7)		20,777 (22.7)	
Female	78,433 (46.1)	7536 (9.6)		13,888 (17.7)	
Age			<0.001		<0.001
19–44 years	24,561 (14.5)	556 (2.3)		657 (2.7)	
45–64 years	75,738 (44.6)	8724 (11.5)		12,227 (16.1)	
65–74 years	43,762 (25.7)	8033 (18.4)		13,085 (29.9)	
≥75 years	25,898 (15.2)	4580 (17.7)		8696 (33.6)	
Treatment outcome			<0.001		0.734
Improved	150,868 (88.8)	19,032 (12.6)		30,723 (20.4)	
Not improved	8191 (4.8)	1133 (13.8)		1705 (20.8)	
Death	10,311 (6.1)	1630 (15.8)		2112 (20.5)	
Other	589 (0.3)	98 (16.6)		125 (21.2)	
Surgery			<0.001		<0.001
Yes	69,202 (40.7)	6574 (9.5)		12,802 (18.5)	
No	100,757 (59.3)	15,319 (15.2)		21,863 (21.7)	
Chemotherapy			<0.001		0.163
Yes	19,345 (11.4)	2750 (14.2)		3872 (20.0)	
No	150,614 (88.6)	19,143 (12.7)		30,793 (20.4)	
Radiotherapy			0.247		0.686
Yes	719 (0.4)	103 (14.3)		151 (21.0)	
No	169,240 (99.6)	21,790 (12.9)		34,514 (20.4)	
Hyperthyroidism			<0.001		0.012
Yes	301 (0.2)	61 (20.3)		79 (26.2)	
No	169,658 (99.8)	21,832 (12.9)		34,586 (20.4)	
Hypothyroidism			<0.001		<0.001
1802 (1.1)	307 (17.0)		476 (26.4)	
No	168,157 (98.9)	21,586 (12.8)		34,189 (20.3)	
Total	169,959 (100.0)	21,893 (12.9)		34,665 (20.4)	

*p*-value is the Bonferroni correction for a χ^2^ test.

**Table 2 jpm-12-01099-t002:** Prevalence of diabetes mellitus and hypertension among patients with major cancers.

Cancer	Total *n* (%)	with Diabetes Mellitus		with Hypertension	
		*n* (%)	*p*-Value	*n* (%)	*p*-Value
Total	169,959 (100.0)	21,893 (12.9)		34,665 (20.4)	
Pancreatic cancer	5514 (3.2)	1649 (29.9)	<0.001	1552 (28.1)	<0.001
Bile duct cancer	3361 (2.0)	763 (22.7)	<0.001	1056 (31.4)	<0.001
Lung cancer	21,227 (12.5)	3260(15.4)	<0.001	5541 (26.1)	<0.001
Gallbladder cancer	1989 (1.2)	308 (15.5)	<0.001	531 (26.7)	<0.001
Small intestine cancer	646 (0.4)	100 (15.5)	0.048	146 (22.6)	0.164
Multiple myeloma	1894 (1.1)	285 (15.0)	0.005	496 (26.2)	<0.001
Kidney cancer	2895 (1.7)	444 (15.3)	<0.001	854 (29.5)	<0.001
Colorectal cancer	21,043 (12.4)	2829 (13.4)	0.009	4861 (23.1)	<0.001
Ureter cancer	620 (0.4)	91 (14.7)	0.181	189 (30.5)	<0.001
Bladder cancer	4489 (2.6)	643 (14.3)	0.003	1233 (27.5)	<0.001
Laryngeal cancer	1197 (0.7)	153 (12.8)	0.918	245 (20.5)	0.951
Liver cancer	19,924 (11.7)	4254 (21.4)	<0.001	4534 (22.8)	<0.001
Stomach cancer	22,198 (13.1)	2800 (12.6)	0.202	4854 (21.9)	<0.001
Prostate cancer	5509 (3.2)	674 (12.2)	0.145	1477 (26.8)	<0.001
Anal cancer	184 (0.1)	22 (12.0)	0.708	43 (23.4)	0.317
Esophageal cancer	2224 (1.3)	258 (11.6)	0.070	490 (22.0)	0.054
Other	55,045 (32.4)	3360 (6.1)	<0.001	6563 (11.9)	<0.001

*p*-value is the Bonferroni correction for a χ^2^ test.

**Table 3 jpm-12-01099-t003:** Association rule mining between cancer disease, diabetes mellitus, and hypertension.

No	Rules	Count	Support	Confidence	Lift	IS Scale
1	(DM)→(HTN)	11,740	0.069	0.536	2.629	0.426
2	(HTN)→(DM)	11,740	0.069	0.339	2.629	0.426
3	(Lung cancer)→(HTN)	5541	0.033	0.261	1.280	0.204
4	(HTN)→(Lung cancer)	5541	0.033	0.160	1.280	0.204
5	(Liver cancer)→(DM)	4254	0.025	0.214	1.658	0.204
6	(DM)→(Liver cancer)	4254	0.025	0.194	1.658	0.204
7	(HTN, Liver cancer)→(DM)	1964	0.012	0.433	3.363	0.197
8	(Colorectal cancer)→(HTN)	4861	0.029	0.231	1.133	0.180
9	(HTN)→(Colorectal cancer)	4861	0.029	0.140	1.133	0.180
10	(Stomach cancer)→(HTN)	4854	0.029	0.219	1.072	0.175
11	(HTN)→(Stomach cancer)	4854	0.029	0.140	1.072	0.175
12	(Liver cancer)→(HTN)	4534	0.027	0.228	1.116	0.173
13	(HTN)→(Liver cancer)	4534	0.027	0.131	1.116	0.173
14	(DM, Lung cancer)→(HTN)	1813	0.011	0.556	2.727	0.171
15	(HTN, Lung cancer)→(DM)	1813	0.011	0.327	2.540	0.165
16	(DM, Liver cancer)→(HTN)	1964	0.012	0.462	2.264	0.162
17	(Lung cancer)→(DM)	3260	0.019	0.154	1.192	0.151
18	(DM)→(Lung cancer)	3260	0.019	0.149	1.192	0.151
19	(Colorectal cancer)→(DM)	2829	0.017	0.134	1.044	0.132
20	(DM)→(Colorectal cancer)	2829	0.017	0.129	1.044	0.132
21	(DM, HTN)→(Liver cancer)	1964	0.012	0.167	1.427	0.128
22	(DM, HTN)→(Lung cancer)	1813	0.011	0.154	1.236	0.115

Abbreviations: DM, diabetes mellitus; HTN, hypertension; IS, interest support.

**Table 4 jpm-12-01099-t004:** Logistic regression analysis of mortality factors according to cancer.

Variables	B	S.E.	OR	95% CI	*p*-Value
All cancer (*n* = 169,959)					
Diabetes mellitus	−0.086	0.030	0.917	(0.865–0.973)	0.004
Hypertension	0.182	0.027	1.200	(1.137–1.266)	<0.001
Hypothyroidism	0.033	0.101	1.033	(0.849–1.259)	0.743
Hyperthyroidism	0.568	0.325	1.765	(0.933–3.340)	0.081
Stomach cancer (*n* = 22,198)					
Diabetes mellitus	−0.059	0.097	0.943	(0.780–1.140)	0.543
Hypertension	0.402	0.087	1.495	(1.261–1.772)	<0.001
Hypothyroidism	−0.190	0.355	0.827	(0.412–1.657)	0.591
Hyperthyroidism	0.629	1.024	1.876	(0.252–13.966)	0.539
Colorectal cancer (*n* = 21,043)					
Diabetes mellitus	0.060	0.104	1.061	(0.865–1.302)	0.569
Hypertension	0.182	0.088	1.199	(1.008–1.426)	0.040
Hypothyroidism	−0.137	0.399	0.872	(0.399–1.907)	0.731
Hyperthyroidism	0.236	1.033	1.266	(0.167–9.588)	0.819
Liver cancer (*n* = 19,924)					
Diabetes mellitus	0.235	0.065	1.265	(1.113–1.437)	<0.001
Hypertension	0.217	0.065	1.242	(1.093–1.411)	0.001
Hypothyroidism	−0.724	0.236	0.485	(0.305–0.771)	0.002
Hyperthyroidism	-	-	-	-	-
Lung cancer (*n* = 21,227)					
Diabetes mellitus	−0.079	0.066	0.924	(0.812–1.051)	0.228
Hypertension	0.239	0.057	1.270	(1.135–1.421)	<0.001
Hypothyroidism	−0.249	0.242	0.779	(0.485–1.252)	0.303
Hyperthyroidism	0.011	0.533	1.011	(0.356–2.872)	0.984

Control variables: sex, age, operation, chemotherapy, radiotherapy. Abbreviations: OR, odds ratio; CI, confidence interval.

## Data Availability

Restrictions apply to the availability of these data. Data were obtained from KDCA and are available from https://www.kdca.go.kr/contents.es?mid=a20303010502 (accessed on 8 March 2020).

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
