# Peer review of "The Impact of the Association between Cancer and Diabetes Mellitus on Mortality"

_jpm, 2022, doi:10.3390/jpm12071099_

Round 1

Reviewer 1 Report

Dear authors, I've appreciated a lot your paper

this topic is of great interest taking into account the incredible prevalence of diabetes in the entire world population

i was captured by the association between DM and pancreatic cancer,

therefore I have only one recommendation to introduce the concept that the gestational diabetes, not only DM associate with a higher risk of pancreatic cancer as demonstrated by this two papers

that I recommend to read and cite

PMID: 34454160.

PMID: 17705823

Reviewer 2 Report

The manuscript entitled “The impact of mortality on the association between cancer and diabetes mellitus” submitted by Kim and Kim evaluated the effect of the association between cancer and diabetes mellitus. The authors analyzed the data of 169,959 cancer patients, and the outcome is outstanding.  However, some modifications are required from the authors’ side to improve the quality of the manuscript.

11.      The title failed to reflect the outcome, and it should be “The impact of the association between cancer and diabetes mellitus on mortality”.

22.  Line 237: Authors mentioned that “Hyperthyroidism and hypothyroidism had no impact  237   on the cancer mortality rates”. But from table 4, I found that hypothyroidism reduced the mortality rate 0.485 times (p=0.002).

33. The level of significance (p-value) never be zero; it may be p<0.001, p<0.00001 etc. Authors should revise the p-values in table 4 and in the text.

44.  Line 190, ‘liver’ should be replaced with ‘liver cancer’.

55. The introduction section should be improved by explaining the interlink between the four types of cancer and diabetes mellitus.

66. Bonferroni correction of p-values should be performed.

77. Some grammatical errors were found that should be corrected.

Reviewer 3 Report

The research idea is interesting. The research can be extended by applying the methodology of survival studies and establishing to what extent the associated pathology (diabets, cardiovasculare disease) can influence the prognosis of the cancer patient.
